# High-speed laser writing of structural colors for full-color inkless printing

Jiao Geng[1,2], Liye Xu[1,2], Wei Yan[1,2], Liping Shi ®[1,2] ✉ & Min Qiu ®[1,2] ✉

It is a formidable challenge to simultaneously achieve wide-gamut, high-resolution, high-speed while low-cost manufacturability, long-term stability, and viewing-angle independence in structural colors for practical applications. The conventional nanofabrication techniques fail to match the requirement in low-cost, large-scale and flexible manufacturing. Processing by pulsed lasers can achieve high throughput while suffering from a narrow gamut of ~15% sRGB or angle-dependent colors. Here, we demonstrate an all-in-one solution for ultrafast laser-produced structural colors on ultrathin hybrid films that comprise an absorbent dielectric TiAlN layer coating on a metallic TiN layer. Under laser irradiation, the absorption behaviours of the TiAlN-TiN hybrid films are tailored by photothermal-induced oxidation on the topmost TiAlN. The oxidized films exhibit double-resonance absorption, which is due to the non-trivial phase shifts both at the oxide-TiAlN interface, and at the TiAlN-TiN interface. By varying the accumulated laser fluence to modulate the oxidation depth, angle-robust structural colors with unprecedented large-gamut of ~90% sRGB are obtained. The highest printing speed reaches 10 cm²/s and the highest resolution exceeds 10000 dpi. The durability of the laser-printed colors is confirmed by fastness examination, including salt spray, double-85, light bleaching, and adhesion tests. These features render our technique to be competitive for industrial applications.

Inkjet or laser printers rely on ink or toner cartridges, in which colorful pigments serve to selectively absorb visible light within a spectral range. However, the conventional pigments are toxic and environment unfriendly. Further, their long-term stability in ambient conditions is generally poor, because the organic pigments tend to degrade over time and thus losing their chroma as well as brightness[1]. Therefore, the development of structural colors-based ink-free printers is demanded on looming. In recent years, the structural colors[2] arising from light scattering, absorption, diffraction or interference by micro/nanostructures including plasmonic[3–10] and all-dielectric metasurfaces[11–21], diffractive elements[22,23], microfibrillation and photonic crystals[24,25], Fabry–Perot (FP) cavities[26–31], multilayered dielectric films[32–34], lossy dielectrics on metallic coatings[35], Fano-resonance optical coatings[36],

are promising to achieve pigments-free colorful printing. It is hence becoming an attractive technique in many applications, such as the labeling of serial number, barcode, quick response code, company logo, trade marker, anti-counterfeiting, to name a few. However, surface coloring in terms of the conventional nanofabrication techniques, such as electron beam lithography[37,38], focused ion beam milling[39] and nanoimprinting lithography[40], is facing the nanoscale and macroscale processing barrier. Production of large-scale colored surfaces with these techniques is incompatible with the demands of low-cost mass manufacturing.

As an alternative, surface coloring by lasers, exhibiting high throughput of >10 mm²/s[41], is appealing to overcome this barrier. In addition, lasers are capable of large-scale fabrication on highly rough

[1]Key Laboratory of 3D Micro/Nano Fabrication and Characterization of Zhejiang Province, School of Engineering, Westlake University, 18 Shilongshan Road, Hangzhou 310024 Zhejiang Province, China. [2]Institute of Advanced Technology, Westlake Institute for Advanced Study, 18 Shilongshan Road, Hangzhou 310024 Zhejiang Province, China. ✉e-mail: shiliping@westlake.edu.cn; qiumin@westlake.edu.cn

surfaces, which are challenging for conventional techniques such as electron beam lithography and nanoimprinting[42]. Laser coloring generally contains three approaches that are based on different mechanisms: plasmonic colors from randomly self-organized metallic nanoparticles[42], diffractive colors from laser-induced periodic surface structures (LIPSS, i.e., nanogratings)[43–45], and interfering colors from thin films including transparent oxide layer[46] and FP-cavities[47,48].

The laser-produced plasmonic colors on noble metals exhibit the advantages of viewing angle-independent but face the problems of narrow gamut (~15% sRGB)[42] and low stability[41]. Passivation coatings are generally required to protect the colors, however, the passivation layer will change the colors because the plasmonic resonances are highly sensitive to the surrounding dielectrics[42,49]. LIPSS exhibit iridescence that strongly limits its practical applications in patterning[50].

Another widely adopted laser coloring technique is in terms of laser-induced oxide layer, generally occurring on metallic surfaces such as titanium alloy or stainless steel[41,46,51]. This approach exhibits the advantages of high stability and productivity. In this case, colors come from interference of reflection from top and bottom surfaces of the oxide layer. The optical path difference $\delta$ between the two reflected beams can be expressed by $\delta = 2hn_1^2 / \sqrt{n_1^2 - (\sin\theta_i)^2}$, where $h$, $n_1$ and $\theta_i$ denote the thickness and refractive index of the oxide film, and the incidence angle, respectively. At normal incidence ($\theta_i = 0$), when $\delta = 2hn_1$ (i.e., $h = \lambda/2n_1$) for a wavelength $\lambda$, the specific color originates from constructive interference-enhanced reflection, the same effect that is responsible for soap bubbles. With the increase of oxide layer thickness, the color coordinates only move in a clockwise direction following an elliptic curve[41,46,52], but green colors are difficult to be achieved and thus its gamut is also narrow (~35% sRGB)[41]. This has been attributed to the film thickness, which requires large enough for the origin of these colors associated with long-wavelength[52]. Furthermore, the colors from laser-induced oxide layer are dependent on viewing angles, which is disadvantageous for practical applications[41,53,54].

The structural colors originating from FP-cavities can be insensitive to the viewing angles[55,56]. The FP-cavities consist of two metallic layers that are separated by a lossless dielectric spacer layer. The reflected colors are strongly dependent on the thickness of dielectric layer and weakly sensitive to the thickness of the topmost metallic film. However, it is challenging to directly modify the spacer layer by lasers because of the high reflection of the metallic film. As an alternative, laser-induced polymer-assisted photochemical metal deposition (PPD) has been employed to control the thickness of the topmost metallic layer[27]. Nevertheless, this technique is also facing the problem of a narrow gamut (~25% sRGB). Further, both plasmonic colors and FP-cavities rely on noble metals such as Au, Ag and Cu, exhibiting low wear resistance and thus poor abrasion stability[27,41,47].

Here, we demonstrate a laser-coloring scheme to solve the aforementioned problems. Structural colors with wide gamut, durable, large-scale and viewing angle-insensitive—key parameters that are decisive for the adoption in decoration technologies[57]—are simultaneously achieved via laser-induced oxidation on TiAlN–TiN hybrid films.

## Results and discussion
### Mechanisms of laser coloring on hybrid films

As illustrated in Fig. 1, highly absorbing dielectric TiAlN, coating on a reflective TiN film[58], serves as "inorganic ink" for laser color printing. The refractive indices of TiAlN and TiN are given in Supplementary Fig. 1. Such bilayer films, acting as a broadband absorber[59–62], represent thickness-dependent colors, as displayed by the photograph in Fig. 2a. However, its gamut is rather narrow, as shown in International Commission on Illumination (CIE) 1931 $x$–$y$ chromaticity diagram (blue curve in Fig. 2b). But interestingly, when depositing an ultrathin transparent dielectric such as AlN or $Al_2O_3$ film onto the TiAlN–TiN absorber, the observed colors can be dramatically changed. Especially, red and green colors are acquired. The corresponding gamut is obviously extended, as shown by the numerical simulations in Fig. 2b.

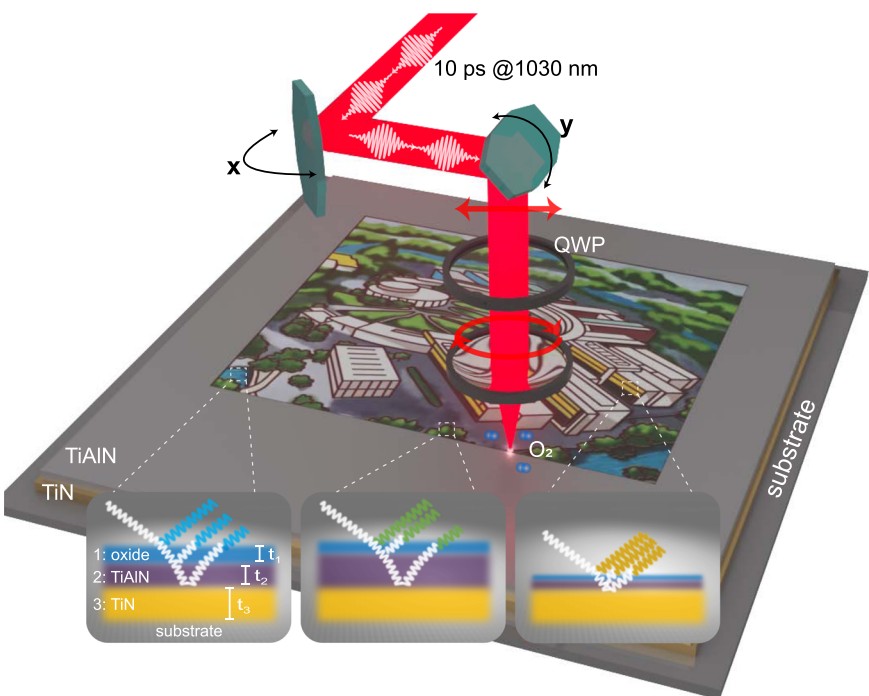

**Fig. 1 | Scheme of surface coloring by ultrafast laser.** The structural colors originate from laser-induced oxide layer on highly absorbing while hard ceramic titanium aluminum nitride (TiAlN) film. The titanium nitride (TiN) film behaves as a reflector. QWP: quarter-wave plate. Large-scale printing is achieved by meander scanning. The interference of light that transmits and reflects at the interfaces gives rise to strongly resonant absorption and displays tunable colors. $t_1$: thickness of laser-induced oxide layer (medium 1); $t_2$: thickness of remaining TiAlN film (medium 2); $t_3$: thickness of TiN (medium 3). Laser simultaneously changes $t_1$ and $t_2$, which results in various reflected colors (insets).

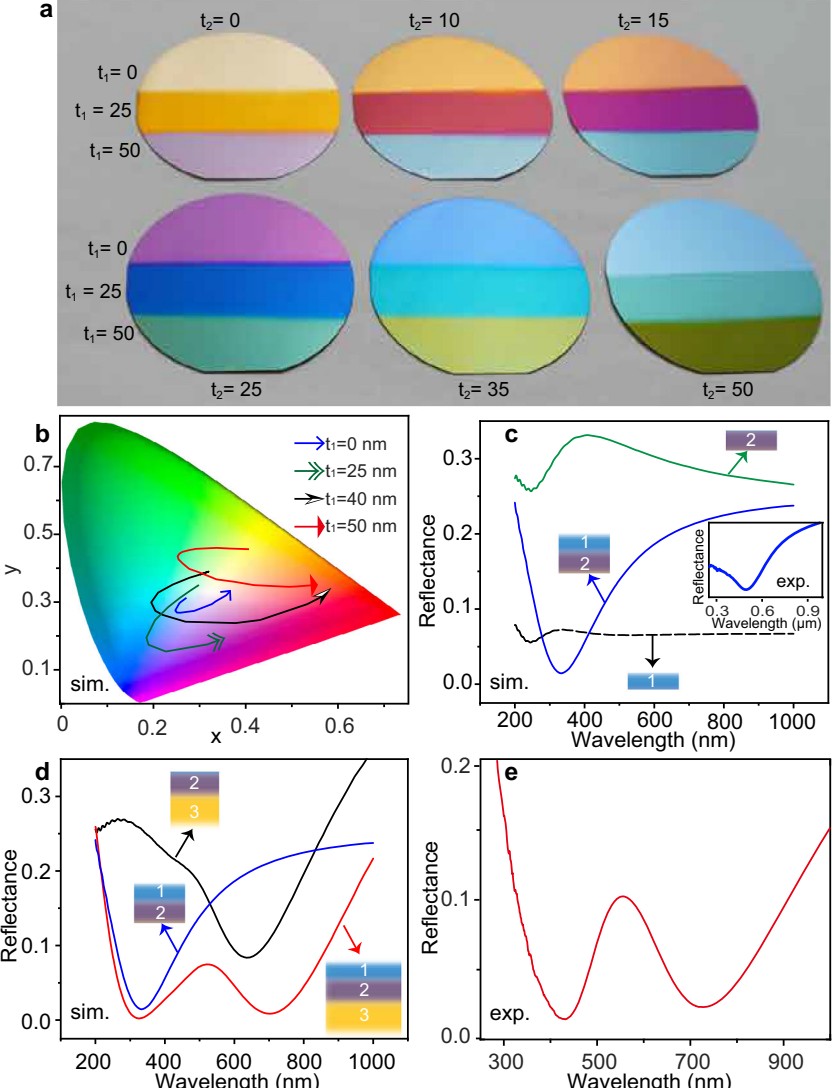

**Fig. 2 | Optical properties of TiAlN–TiN hybrid films. a** Photograph of surface coloring with various thickness of AlN ($t_1$) coating subsequent with different thickness of TiAlN coatings ($t_2$ = 0, 10, 15, 25, 35, 50 in unit of nm) on $t_3$ = 50 nm TiN. The substrates are 2-in. Si wafers. On each sample, three different thickness of AlN are employed: top $t_1$ = 0; middle $t_1$ = 25 nm; bottom $t_1$ = 50 nm. Please note that masks were used to change the thickness of AlN in different areas during film coatings. **b** Numerically simulated CIE 1931color coordinates for $t_1$ = 25 nm AlN or $t_1$ = 40 nm and 50 nm Al$_2$O$_3$ coating on variable thickness of TiAlN ($t_2$), which decreases from 50 to 10 nm along the arrows. **c** Simulation of normal-incidence reflectance spectra of Al$_2$O$_3$, TiAlN, and their stack coating, respectively. The

thickness of each coating is 40 nm. Inset: measured reflectance spectrum of 40 nm AlN coating on 40 nm TiAlN. **d** Simulated reflectance spectra at normal incidence of Al$_2$O$_3$-on-TiAlN (blue curve), TiAlN-on-TiN (black curve), and Al$_2$O$_3$-on-TiAlN-on-TiN (red curve), respectively. The thickness of Al$_2$O$_3$ (medium 1), TiAlN (medium 2), and TiN (medium 3) are 35, 40, and 50 nm, respectively. **e** Measured reflection spectrum of a sample with $t_{1,2,3}$ = 50 nm. Please note that in experiments, the lossless dielectric coating (medium 1) **a, c, e** is AlN, because it is more convenient to use the same targets as that for TiAlN during the sputtering process, while in simulations, **c, d**, medium 1 is set as Al$_2$O$_3$ to match the laser-produced oxide layer on TiAlN. Source data is available.

Changing structural colors of the thin film absorber by a transparent dielectric coating can be understood from the Snell's law[63]. At incidence angle of $\theta_i$ and refraction angle of $\theta_r$ the reflection coefficient at an interface is

$$\tilde{r}_{m,n} = \frac{\tilde{n}_m\cos(\theta_i) - \tilde{n}_n\cos(\theta_r)}{\tilde{n}_m\cos(\theta_i) + \tilde{n}_n\cos(\theta_r)} \qquad (1)$$

where $\tilde{n}_{m,n} = n_{m,n} + ik_{m,n}$ is the complex-valued refractive index of medium $m$ and $n$, with {$m, n$} = {1, 2} or {2, 3}. Assuming oxide layer has a refractive index close to Al$_2$O$_3$ (medium 1, $h_1$ = 35 nm, $\tilde{n}_1$ = 1.75, the stacking of medium 1 onto medium 2 (TiAlN: $\tilde{n}_2$ = 2.6 + 1.05$i$ exists a resonance absorption at 330 nm in case of normal incidence ($\theta_{i,r}$ = 0),

as numerically simulated in Fig. 2c. According to Eq. (1), the reflection at the interface between them has a nontrivial phase shift $\Phi_{1,2} = \tan^{-1}\left[\frac{\mathrm{Im}(\tilde{r}_{1,2})}{\mathrm{Re}(\tilde{r}_{1,2})}\right] = 1.2\pi$. The total phase shift is $\delta\varphi = \Phi_{1,2} + \left(2h_1n_1\frac{2\pi}{\lambda} - \pi\right) = \pi$. Therefore, the corresponding colors are caused by destructive interference-induced resonance absorption, differing from the mechanisms of soap bubbles that are caused by constructive interference as mentioned above. Likewise, the non-trivial phase shift between medium 2 and medium 3 also results in a resonance absorption at ~600 nm ($\Phi_{2,3}$ = 1.4$\pi$, as shown in Fig. 2d. The joint behaviors of these three layers leads to a double resonance absorption (Fig. 2d), which is confirmed by the experimental result in Fig. 2e. Such a double resonance offers more freedom to tune the

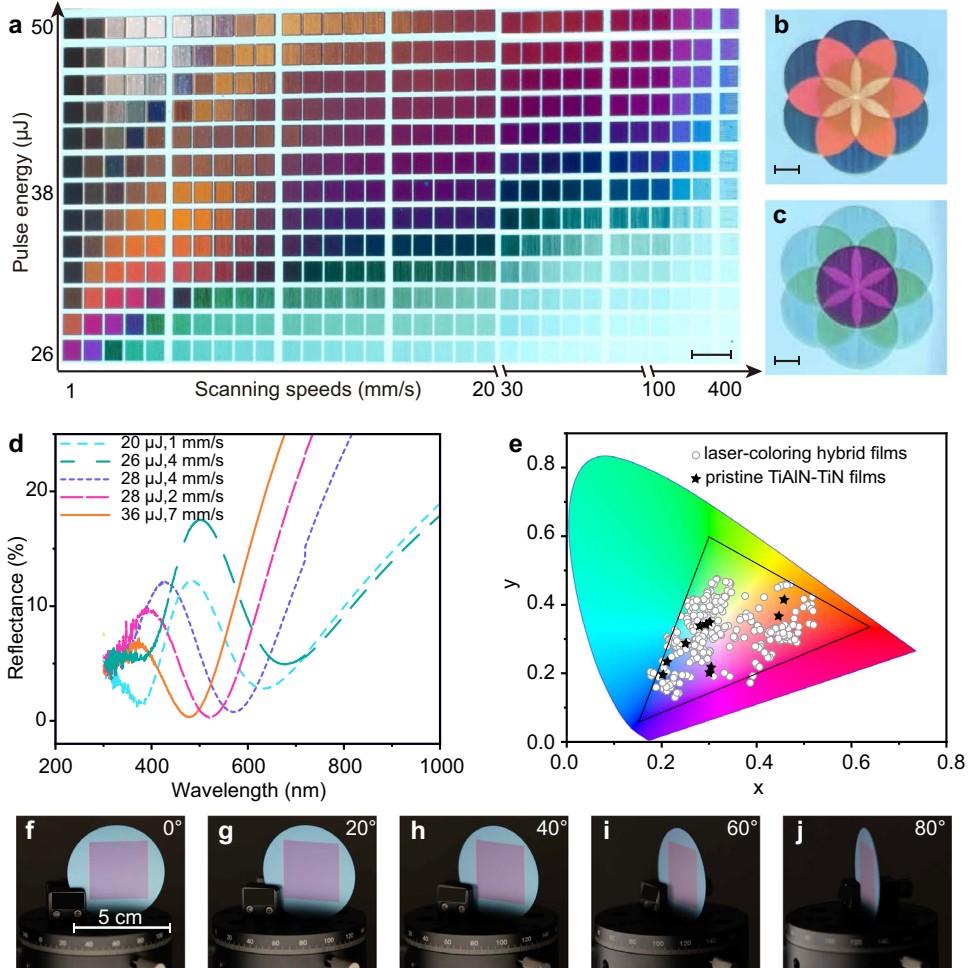

**Fig. 3 | Optical properties of laser-printed colors on TiAlN–TiN hybrid films.**
**a** Matrix palettes produced with various pulse energies and scanning speeds. The thickness of TiAlN and TiN are 60 nm and 50 nm, respectively. **b**, **c** Seven circles that are independently written. The colors in the overlapping areas are changed, indicating that the colors are rewritable. The scale bars are 5 mm. **d** Near-normal

incidence (6°) reflectance spectra of several representative laser-written colors. **e** Experimentally measured CIE chromaticity diagram, comparing the colors of pristine bilayer TiAlN–TiN films (stars) and laser-produced colors (circles). **f–j** Photographs of a laser-written color swatch taken at five tilted angles. Source data is available.

optical absorption and thus resulting in a wider gamut than the original TiAlN–TiN absorber. More importantly, as these coatings are much thinner than the visible light wavelength, the phase accumulation due to the propagation through the film is small. Therefore, they display viewing angles-insensitive colors.

## Spectral analysis of laser-written structural colors

In experiments, the lossless dielectric layer is formed through laser-induced oxidation on the surface of TiAlN. As shown in Fig. 1, a linearly polarized incident picosecond laser is rotated by a quarter-wave plate to circular polarization. This is to avoid the unwanted iridescence that comes from LIPSS-induced light diffraction (see Supplementary Fig. 2)[64]. Figure 3a shows a matrix palette that is produced on 60-nm-TiAlN coating on 50-nm-TiN by meander scanning with a line spacing of $L_s = 20\,\mu m$ between each successive line. The applied laser repetition rate is $f = 5\,kHz$, and the laser spot diameter is $\sigma = 120\,\mu m$. Various vibrant colors, spanning from red, orange, yellow, green, blue to purple, are observed. The colors are dependent on the scanning speeds ($v$ ranging from 1 to 400 mm/s) and laser pulse energy ($E$, spanning from 26 to 50 μJ). These parameters define the total accumulated laser fluence $F_t = \frac{2E}{\pi\sigma^2}N_{eff}$, with $N_{eff} = \frac{f\sigma}{v}$ denoting the effective number of irradiated pulses. Nevertheless, the much larger ranges of $v$ (or $N_{eff}$) with respect to that of $E$

suggests that the oxidation process is more sensitive to the latter. This can be attributed to that the oxidation depth does not linearly increase with $N_{eff}$, because the penetration of $O_2$ through the formed oxide layer decreases exponentially, decelerating and eventually halting the growth process. Interestingly, the generated colors can be locally retouched by a repeated scanning, which changes $F_t$, as shown in Fig. 3b, c.

When using a high repetition rate of $f = 175\,kHz$, the printing throughput ($\eta = L_s v$) can be improved to $\eta = 10\,cm^2/s$ with $v = 10\,m/s$ and $L_s = 100\,\mu m$, as shown in Supplementary Movie 1. The printing resolution is essentially limited to the beam diameter of laser spot, which is tested by producing colors with a single shot and line scanning (see Supplementary Fig. 3a, b). The diameter of a typical printing spot upon our laser-marking system is ~70 μm. Therefore, the spatial resolution is evaluated to be 2.54 cm/70 μm = 360 dpi, which is acceptable for most practical applications. The resolution can be improved by utilizing a smaller laser beam spot. For instance, when focusing the laser beam by an objective lens (NA = 0.9), the printing resolution exceeds 10,000 dpi (see Supplementary Fig. 3c–f). The spot sizes of pigments are generally on the order of 25 μm, resulting in a resolution ~1000 dpi. Therefore, our laser-printing ink-free structural colors can have one order of magnitude higher resolution than the conventional printers.

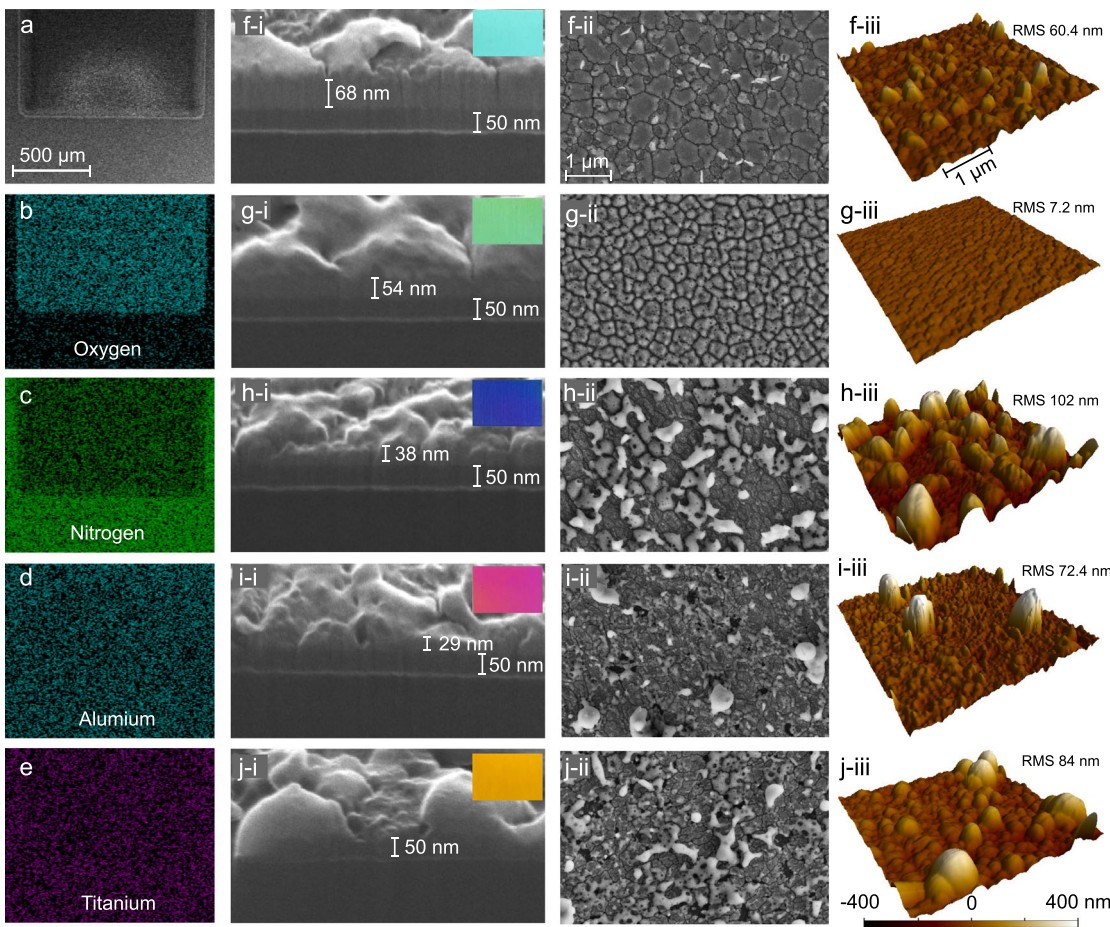

**Fig. 4 | Material properties of laser-printed colors. a–e** SEM image and corresponding EDX maps of a laser-written area with green color. Cross-sectional view images that are obtained by focused ion beam milling [(**f-i**)–(**j-i**)], high-resolution SEM images [(**f-ii**)-(**j-ii**)] and AFM images [(**f-iii**)-(**j-iii**)] of five different laser-colored areas (light blue in **f**; green in **g**; blue in **h**; red in **i**; and yellow in **j**) that are obtained by different scanning speeds or pulse energies. Please note that the RMS are measured by AFM in 40 μm × 40 μm areas. The pulse energy and scanning speed from **f** to **j** are 21, 28, 28, 28, 42 μJ, and 1, 7, 4, 2, 9 mm/s, respectively. The insets show their corresponding color swatches.

The reflectance spectra of several representative colors are plotted in Fig. 3d. Indeed, we find that the laser-treated films exhibit double-resonance absorption, in agreement with the simulation in Fig. 2d and the intentionally sputtered three-layer film in Fig. 2e. Plotting the reflectance spectra of the matrix palette into CIE 1931 chromaticity diagram, as shown in Fig. 3e, we verify that the laser-printed colors have a wide gamut (~90% sRGB), which is obviously broader than that of the pristine TiAlN–TiN bilayer. The reflectance spectra versus incidence angles are plotted in Supplementary Fig. 4. The spectral profiles and the reflected peaks are nearly unchanged, suggesting that the colors are rather insensitive to the viewing angles. This is confirmed by photographs of a color swatch viewing at various titled angles, as shown in Fig. 3f–j. Therefore, compared to structural colors from laser-induced oxide layer on stainless steels[41], our coloring on lossy dielectric exhibits the advantages of viewing angle independence and wider gamut.

## Surface material analysis

In order to verify that the surface coloring indeed arises from laser-induced oxidation, we perform extensive surface material analysis by energy dispersive X-ray spectroscopy (EDX), focused ion beam (FIB) milling, atomic force microscopy (AFM), X-ray diffraction (XRD), and x-ray photoelectron spectroscopy (XPS). Figure 4a–e list the SEM image of a large area of green color and corresponding two-dimensional EDX maps of oxygen, nitrogen, titanium and aluminum, respectively. The EDX maps confirm that, indeed, nitrogen has been

partially replaced by oxygen, while the components titanium and aluminum are nearly unchanged. The EDX maps of other laser-written colors are shown in Supplementary Fig. 5, which further confirm that all generated colors are relevant to oxidation of TiAlN.

Next, the colored films are milled by a Ga⁺-FIB to observe their cross-sectional views, as depicted in Fig. 4f-i, j-i. One can see that, for light-blue color that is obtained at low irradiance (inset in Fig. 3f), the oxidation degree is rather weak. Its high-resolution SEM image in Fig. 4f-ii reveals that the laser-oxidized layer exhibits multiple cracks, with a surface roughness (RMS) of 60.4 nm as measured by AFM [Fig. 4f-iii]. The formation of cracks may be attributed to the temperature gradient and thermal stress[65]. At medium power while high scanning speed, the color becomes green (inset in Fig. 4g) and three layers are clearly observed, as shown in Fig. 4g-i, which is in accordance with the numerical simulations in Fig. 2d. The surface cracks become smaller, and the roughness is very small in this case (RMS = 7.2 nm), suggesting that the oxide cracks may have been reorganized under laser irradiation. For dark blue, red and yellow colors that are obtained at high accumulated fluence, the thickness of oxidation and remaining TiAlN layers decreases with the increase of fluence. This indicates that the oxide layer has been partially ablated, which is further confirmed by the EDX measurement in Supplementary Fig. 5. Some ablative debris in form of aluminum oxide (high-resolution EDX maps in Supplementary Fig. 6) is randomly redeposited on the surface, resulting in a slightly higher roughness, as shown by the SEM images in Fig. 4h-ii, j-ii and AFM images in Fig. 4h-iii, j-iii. As seen from the

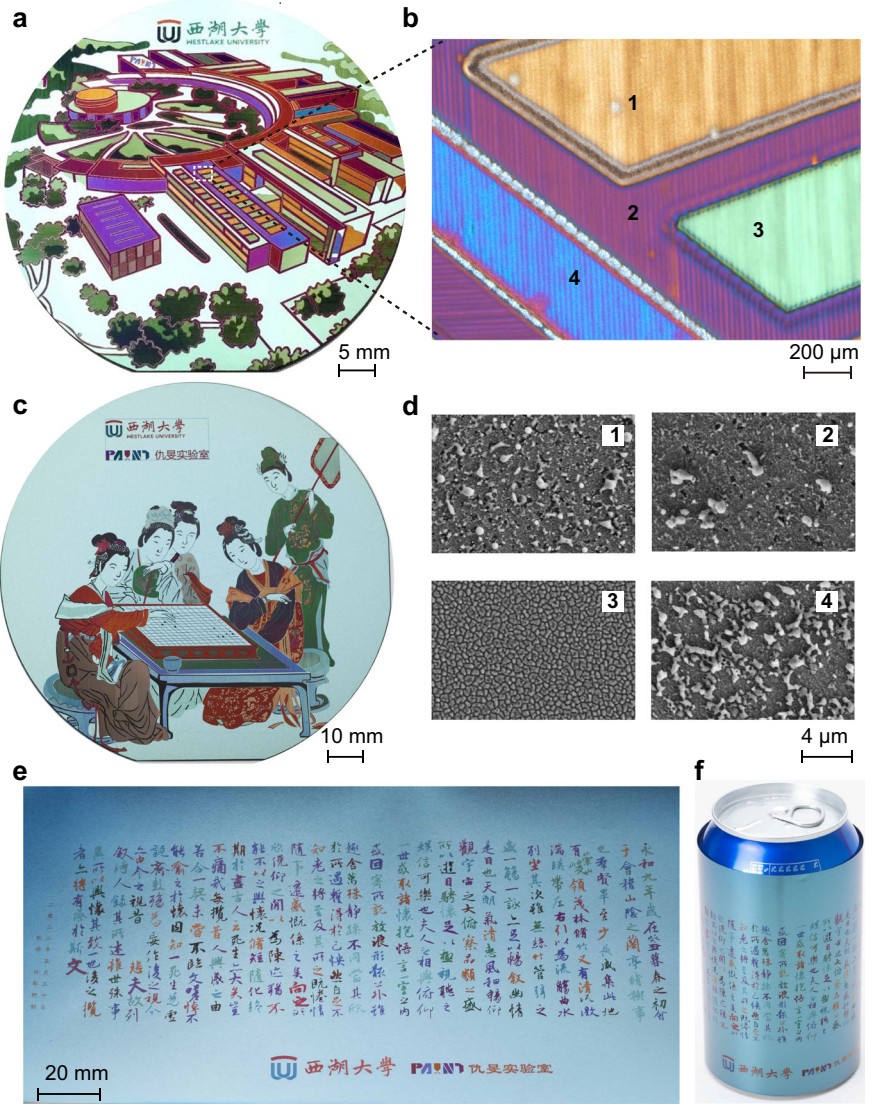

**Fig. 5 | Large-scale laser color printing. a** Photograph of a structural color-based large-scale pattern ("The academic ring of Yungu campus at Westlake university", with the permission from Westlake university), which is printed on TiAlN–TiN hybrid films on a polished 2-in. Si wafer. **b** Optical microscopy image and corresponding SEM images in areas (**d**, 1–4). Photographs of laser-printed large-scale patterns on different substrates, including unpolished backside of a 4-in. Si wafer in (**c**) ("Spring Dawn in the Han Palace", created by Ying Qiu in the Ming Dynasty, 1498–1552), and on a 50-μm-thick stainless-steel foil in (**e**) ("Orchid Pavilion Collection", created by Xizhi Wang in the Tsin Dynasty, 303–361). The thickness of TiAlN and TiN are 60 nm and 50 nm, respectively. **f** The same pattern as that in (**e**), while has been rolled onto a drink can.

reflectance spectra in Supplementary Fig. 7, unlike the metallic nanoparticles that usually exist plasmonic resonances, the dielectric alumina nanoparticles do not cause additional resonance absorption, and thus the spectral profiles of the multilayer hybrid films are not obviously changed. In addition, XRD and XPS are also employed to investigate the laser-modified material properties (see Supplementary Fig. 8), which further support the mechanisms of laser-induced oxidation for coloring on our hybrid films.

**Large-scale laser printing**

To illustrate the capability of printing large-scale colorful images by laser-written structural colors, we produce several photographs on different substrates. TiN film with a thickness of 50 nm and TiAlN film with a thickness of 60 nm are subsequently deposited onto the substrates by RF magnetron reactive sputtering. Figure 5a exhibits a printed "The academic ring of Yungu campus at Westlake university" on a polished single-crystalline Si wafer. The low-magnification optical microscopy and corresponding SEM images are shown in Fig. 5b, d, respectively.

One can see from the high-resolution SEM images in Fig. 5d that there exist random nanoparticles on laser-induced cracked surfaces, which are in agreement with the analysis of color swatches in Fig. 4.

Next, we also carry out printing on an unpolished backside of a 4-in. Si wafer that has high topographic relief, as shown in Fig. 5c. More interestingly, we find that compared to polished surfaces, the use of rough surfaces as substrates delivers more uniform brightness versus viewing angles. The reason is that when coloring on polished surfaces, the visualized colors mainly come from specular reflection, while on rough surfaces the colors are from diffuse reflection. Our technique also works on flexible substrates. As an example, 50-nm-TiN and 60-nm-TiAlN hybrid films are successively deposited on a 50-nm-thick stainless-steel foil. As shown in Fig. 5e, a famous Chinese calligraphy "Orchid Pavilion Collection" is printed in a $20 \times 5$ cm$^2$ area in 35 s. Such flexible foil can be rolled up onto a beverage can, as shown in Fig. 5f. The printing processes are highly reproducible. We fabricate 64 identical samples (see Supplementary Fig. 9). The difference among them is nearly not perceptible by the human eye.

## Aging tests

In practical applications, the durability of structural colors, for example, in terms of resistance to bleaching, abrasion and corrosion is also vital for any decoration technology. The aging tests for laser-written structural colors are independently performed through ultraviolet light bleaching, salt fog and high-temperature high-humidity corrosion, and adhesion test (see Methods). After exposing to these extreme environments for 120 h, the adhesion of the structural colors remains to be the highest level, and the caused color differences are fairly small (see Supplementary Fig. 10). This is an acceptable match in commercial reproduction on printing presses that requires $\triangle E^*_{ab}<7$[54]. It should be pointed out that the laser-printed plasmonic colors need to be appropriately embedded or protected by coatings[37,57], but in our case, the laser-produced oxide film already behaves as a protective coating. Indeed, it is because of the oxidative layer that results in the high mechanical and thermal stability of TiAlN for its applications in automotive and aerospace industries[66].

In conclusion, we have demonstrated ultrafast laser coloring on resonant absorbers that are composed of TiAlN on TiN hybrid film with a total thickness of 110 nm. Such ultrathin films not only significantly decrease the material cost and growth time, but also result in small dependence of colors on the incidence angles and run against intuition given our everyday experience with thin film interfering colors. Using the conventional nanofabrication techniques to generate colorful patterns on thin film absorbers generally requires multiple steps of contact photolithography with alignment[59]. However, a single step is sufficient for printing colors by ultrafast lasers. Laser-induced oxidation on the surface of absorbing dielectric films alters the reflectance spectra of the TiAlN–TiN absorbers and thus resulting in oxide and remaining TiAlN thickness-dependent colors. The formation of a transparent oxide layer significantly widens the gamut of the hybrid thin films. Meanwhile, the hard ceramic material TiAlN exhibits excellent resistance in mechanical, thermal, chemical and abrasion. As a result, the laser printed colors on TiAlN-TiN films present wide gamut, wide viewing angles, high-throughput, high-resolution and high-durability. Moreover, the noncontact laser coloring can be applied on highly rough surfaces that are challenging for the conventional large-scale nanofabrication techniques such as nanoimprinting. Therefore, our technique provides a competitive candidate for practical applications of structural colors in decoration technologies (see Supplementary Fig. 11).

## Methods

### Experimental setups

In our experiments, the surface coloring is carried out by a home-built laser marking machine. A 1030 nm laser (Amplitude, Tangerine) with pulse duration of 10 ps, tunable repetition rate as well as pulse energy was focused by a lens (focal length = 20 cm) onto the samples. The incident laser beam diameter was 3 mm, and the spot diameter was measured to be 120 μm by a beam profiler. The samples were exposed to ambient environment. The laser beam was meanderingly scanned by two galvo mirrors for patterning via surface oxidation. The laser polarization was rotated by a quarter-wave plate to be circular, which is in order to avoid the formation of periodic ripples. For high resolution fabrication, the laser beam was coupled into an optical microscope and focused onto samples by an objective lens with NA = 0.9. The samples were installed on a 3D translation stage.

### Sample fabrication and characterization

The TiN films were deposited on silicon wafers by RF magnetron reactive sputtering at 300 °C, power of 600 W, and N$_2$ flow of 14 sccm and Ar of 56 sccm. The TiAlN films were deposited as the same parameters for TiN while the power of Al target is DC 300 W. The AlN films were coated as the same for TiAlN while the Ti target was off. The thickness of our films was measured with a profiler (Stylus). According

to the measured thickness, the permittivities were retrieved with a variable angle spectroscopic ellipsometer (Woollam). The scanning electron images, and energy dispersive X-ray spectra were measured by a field-emission scanning electron microscope (FE-SEM, Carl Zeiss, Gemini450). The x-ray diffraction analysis of the TiAlN-on-TiN coating on silicon wafer was carried out by Bruker D8 Discover.

### Simulation and measurement of reflectance spectra and calculation of chromaticity

The numerical computations were performed by using commercial software finite-difference-time-domain (FDTD) method (Lumerical FDTD solutions software package). The light source was a plane wave spanning from 400 nm to 2000 nm. The top and bottom boundaries were perfectly matched layers (PML), while periodic boundary conditions were applied in $x-y$ plane. One monitor was set 2 μm above the source to acquire the reflection spectra. The permittivities of the materials involved were measured with the aforementioned ellipsometer. The measured reflectance spectra at near-normal incidence were recorded using a Shimadzu UV−VIS−IR spectrophotometer (UV3600Plus + UV2700). The angle-dependent reflectance and transmittance spectra were measured by an Agilent Cary spectrophotometer (Cary 6000 l, UV−Vis−NIR System). The colors in $x-y$ chromaticity diagram were calculated based on the simulated and measured reflectance spectra and the color-matching functions defined by the CIE 1931[67]. The spectral power distribution can be expressed as: $P(\lambda) = I(\lambda)R(\lambda)$, in which $I(\lambda)$ is the radiance spectrum of light source and $R(\lambda)$ is the reflectance spectrum. The values $X$, $Y$, and $Z$ are given by

$$X = \frac{1}{K} \int_\lambda \bar{x}(\lambda)P(\lambda)d\lambda, \tag{2}$$

$$Y = \frac{1}{K} \int_\lambda \bar{y}(\lambda)P(\lambda)d\lambda, \tag{3}$$

$$Z = \frac{1}{K} \int_\lambda \bar{z}(\lambda)P(\lambda)d\lambda \tag{4}$$

where $\bar{x}$, $\bar{y}$, and $\bar{z}$, are the CIE standard observer functions. The integrals in Eqs. (2−4) were calculated over the visible spectral range (from 380 to 780 nm), and $K = \int_\lambda \bar{y}(\lambda)I(\lambda)d\lambda$ is the normalized constant. As a result, the chromaticity coordinates in CIE 1931 were obtained by $x = \frac{X}{X+Y+Z}$, and $y = \frac{Y}{X+Y+Z}$.

### Color fastness tests

The color fastness tests include salt fog test, double-85 test, light bleaching test and adhesion test, which are generally required for printing industry. The salt fog test is performed according to the standard of GB/T 10125-2021 for 120 h. The density of NaCl is 50 g/L with a sedimentation rate of 2 ml/(80 cm²h). The double-85 test is carried out within the standard of GB/T 1740-2007 for 120 h. The temperature and humidity are 85 °C and 85%. The light bleaching test (GB/T 1865-2009) is irradiated by 340 nm UV light with an irradiance of 0.51 W/(m²nm) for 120 h. The temperature and humidity are 65 °C and 50% RH, respectively. The adhesion test is performed by using 3 M tapes according to the standard of GB/T 9286-2021. The color differences ($\triangle E^*_{ab}$) between the pristine and tested samples are measured by an integrating sphere spectrophotometer using the D65 light source.

### Reporting summary

Further information on research design is available in the Nature Portfolio Reporting Summary linked to this article.

## Data availability

All data supporting the findings of this study are available in the article and its Supplementary Information. Source data for the following figures are provided with this paper. Figure 2b–e; Fig. 3d, e; Supplementary Fig. 1; Supplementary Fig. 4; Supplementary Fig. 7; Supplementary Fig. 8. Source data are provided with this paper.

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

## Acknowledgements

Min Qiu was supported by the National Key Research and Development Program of China (2017YFA0205700), the National Natural Science Foundation of China (61927820). Jiao Geng is supported by the National Natural Science Foundation of China (62105269), Zhejiang Province Selected Funding for Postdoctoral Research Projects (ZJ2021044), and China Postdoctoral Science Foundation (2021M702916). Liping Shi is supported by the National Natural Science Foundation of China (12004314), the open project program of Wuhan National Laboratory for optoelectronics (2020WNLOKF004) and Zhejiang Provincial Natural Science Foundation of China under Grant No. Q21A040010. The authors thank the technical support from Center for Micro/Nano Fabrication, from instrumentation and Service Center for Physical Sciences and from Instrumentation and Service Center for Molecular Sciences at Westlake University. We thank Mr. Danyang Zhu and Miss. Shan Wu for taking the photographs of the fabricated samples. We thank Miss. Yingchun Wu for milling the samples by focused ion beam. We thank Miss. Ruiqian Meng, Dr. Zhong Chen and Prof. Dianyi Liu for the assistance in measuring the spectra. We also thank Mr. Xi Mu for coating the TiAlN thin films.

## Author contributions

M.Q. supervised the project. L.-P.S. conceived the experiments. J.G., L.-P.S., and L.-Y. X. constructed the setup and carried out the experiments. L.-P.S. and W.Y. performed the numerical simulations. L.-P.S and J.G. analyzed the data and wrote the paper. All authors contributed to the writing and revision of paper.

## Competing interests

The authors declare no competing interests.
