## [Peer Review File · Nature Communications]

High-speed laser writing of structural colors for full-color inkless printingReviewer #1 (Remarks to the Author):

The authors report the high-speed laser colouring of a multilayer thin film system consisting of a TiAlN layer on top of a TiN layer. The colours are reported to come from laser induced oxidation of the TiAlN layer which results in reflection of incoming visible light at the oxide-TiAlN and TiAlN-TiN interfaces.

The novelty lies in the combination of techniques where laser-induced oxidation (formation of the alumina layer in other paper - see below - was formed using atomic layer deposition or electrodeposition) is used to tune the oxide thickness on top of the TiAlN-TiN system, therefore tuning the resulting colours. In addition, the colours are automatically protected by the alumina coating, as demonstrated by the authors via various durability tests.

The creation of angle independent colours from multilayered systems is however not new. The authors recently published another article showing angle-independent colour formation on the same material system:

J. Geng, L. Shi, J. Ni, Q. Jia, W. Yan, and M. Qiu, "Wear-resistant surface coloring by ultrathin optical coatings," *Photonix*, vol. 3, no. 1, p. 14, Jun. 2022, doi: 10.1186/s43074-022-00061-5.

Other authors have also used multilayered systems to create angle independent colours. C. Yang et al., "Compact Multilayer Film Structure for Angle Insensitive Color Filtering," *Sci Rep*, vol. 5, no. 1, Art. no. 1, Mar. 2015, doi: 10.1038/srep09285.)

K. Feng et al., "All-dielectric thin films based on single silicon materials for angle-insensitive structural colors," *Opt. Lett.*, OL, vol. 46, no. 20, pp. 5161–5164, Oct. 2021, doi:10.1364/OL.441534

Z. Li, S. Butun, and K. Aydin, "Large-Area, Lithography-Free Super Absorbers and Color Filters at Visible Frequencies Using Ultrathin Metallic Films," *ACS Photonics*, vol. 2, no. 2, pp. 183–188, Feb. 2015, doi: 10.1021/ph500410u.

The creation of structural colours using oxidation is not new either, especially the addition of an alumina layer to create/tune colours with a similar gamut:

X. Wang, D. Zhang, H. Zhang, Y. Ma, and J. Z. Jiang, "Tuning color by pore depth of metal-coated porous alumina," *Nanotechnology*, vol. 22, no. 30, p. 305306, Jul. 2011, doi: 10.1088/0957-4484/22/30/305306.

J.-M. Guay et al., "Laser-written colours on silver: optical effect of alumina coating," *Nanophotonics*, vol. 8, no. 5, pp. 807–822, May 2019, doi: 10.1515/nanoph-2018-0202.

While the rendered colours are mostly a result of the complex interference effects, there may be some structural effects are well that should be discussed. It is possible to see a little bit of that surface morphology in Fig. 3(f-j) and the AFM image Fig. 3(k-l) but only for 1 condition. Larger area SEM images or AFM images should be provided for the various conditions shown in Fig. 3(f-j). If surface morphology doesn't affect the Hue, it might at least affect the brightness and thus the resulting colour. This should be discussed in the paper especially when the roughness of larger than the layer thickness.

The authors claim a resolution of 25000dpi based on a spot size of 1µm. However, In Fig. S1, the the area has a diameter closer to 2 or 3 microns. Why not calculate the dpi based on the actual spot size? Furthermore, it would be good to demonstrate laser printing using this spot size on a larger are (like in Fig.2 a) for instance.

In Fig. 2(d), the conditions (parameters) used to create the curves should be written in the Figure caption (e.g. pulse energy, scanning speed, etc.)

Regarding the angle-independence of the colours, the authors should render the colours from the spectra in Fig. 2f to really prove that the colours do not change as small changes in the spectra can lead to observable changes in the rendered colours. The authors show the Hue vs angle to prove angle independence, but the colour appearance

is not only provided by the Hue but also by the brightness, etc. Why not show the colours obtained experimentally as a function of viewing angle?

The authors say "Therefore, compared to laser-written colors directly on metals, [40] our technique exhibits the advantages of viewing angle-independence and wider gamut." suggesting that laser-written colours are not viewing angle independent. However, there are many reports in literature of angle-independent colours based on surface plasmons. (e.g. ref 5 and 41 in the authors list).

When describing Figs. 4a and b, the deposition of the 50-nm-TiN and 50-nm-TiAlN layers on the Si substrates should be mentioned as it is not clear from the text only (even though it is mentioned in the figure caption).

Fig. 1 – "a)" is missing in the Figure caption. In the same figure caption, the comment "Photograph of surface coloring with various TiAlN and AlN coatings on $t_3=50$ nm TiN. The substrates are Si wafers or glasses." is vague and does not allow to identify where the Si or glass substrate were used. Do they make a difference in the colours observed? Why two different substrates were used? This is the only location where glass is mentioned in the text.

Finally, the paper should be proofread to correct the remaining mistakes, discourse errors and writing inconsistencies.

Reviewer #2 (Remarks to the Author):

The manuscript presents laser coloring method to create optical absorbers on TiAlN & TiN hybrid film, for full-color inkless printing use. Comparing to the conventional nanofabrication techniques, the laser method had unique advantages in the cost and growth time, which is very important for practical applications. I found this idea interesting.

1. In introduction, the authors should properly review the previous work in the field and detail what the deficiencies you are targeting at, followed by how their new laser method is working.

2. The scheme in Figure 1a is not very clear. The label of Figure 1a is missed in figure caption. The five rectangles in Fig. 1a are not easy to understand. The colors represent medium 1, 2 & 3 need to be consistent in the picture to avoid possible misunderstandings.

3. The results on the printing speed of 1.4 cm²/s and resolution of 25000 dpi are based on the speculations but not on the results of the real printing.

4. The author said "Plotting the reflectance spectra of the matrix palette into CIE 1931 chromaticity diagram, as shown in Fig. 2(e), we verify that the laser-printed colors have a wide gamut (~90% sRGB)". But how did they come out with the result of "~90% sRGB"? Because the area of the circles in Figure 2e seems to be much smaller than the triangle representing sRGB area. Besides, the "~90% sRGB" is not an excellent performance in the wide gamut in optical meta-surface field, considering many previous works have reached larger gamut than the sRGB area.

5. The data appeared in Figure 2e should be specified clearly if they came from simulation or experiment? Fig. 2g is not seriously based on data and references. The comparison is not scientifically sound and thus it is not suggested to put here.

6. The authors should show the laser-written area in SEM images with different areas and viewing scale to see more units, other than only a very limited area as shown in Fig. 3 and Fig. S1. An overview and clear images showing the quality of the structure created by the laser method is very important for structure color. Continuous or discrete unit structures with small area or large areas, should be created using different

manufacturing parameters. I would like to see some structures in SEM images in the large-scale laser color printing examples in Fig. 4 and Fig. Fig. S3.

7. The durability test on the laser colors was shown in Fig. S4. The photographs on the difference are suggested to see the durability, which was claimed as one of the advantages of this laser method.

8. In Figure 4, the author showed three large-scale laser color printings. However, I had a few questions about this exhibition. First, the author wrote "we find that the generated colors on rough surfaces delivers more uniform brightness than that on polished ones". But the only thing I can tell from Figure 4a&b is that the colors in Figure 4a had a higher saturation than Figure 4b. If the author is talking about the "more uniform brightness" from different observation angles, he should show the photos in different angles and analyze the results. Second, the Figure 4c had a blue gradient on the steel foil, in the top of the picture. Did the gradient come from the substrate itself, like the uneven lighting? In a paper about colors, photo should be taken by the camera with accurate settings to show the original colors of the metasurface. The same issue also existed in Figure S3; the uneven flashlight made the 64 identical samples look different from each other. Finally, I suggest the authors to draw a CIE diagram concluding colors in Figure 4a for comparison, showing the real printing quality of this laser method.

Reviewer #3 (Remarks to the Author):

Geng et al. demonstrated the use of ultrafast lasers to directly write large-area (wafer-scale) and wide-gamut (~90% sRGB) structural colors that are insensitive to viewing angles. The printing speed reaches to $1.4 \text{ cm}^2/\text{second}$, which is impressive, especially compared to the conventional nanofabrication techniques such as electron beam lithography and focused ion beam milling. Furthermore, it should be pointed that although surface coloring by pulsed lasers-induced oxidation has been previously demonstrated on bulky materials such as stainless steel and silicon, nevertheless, this technique is facing the stubborn problems of narrow gamut and viewing angles-dependent colors. In this work, the authors have solved these problems by utilizing TiN-TiAlN hybrid films as 'inorganic ink'. Another big advantage of such hybrid films is that these materials exhibit extremely high hardness and stability. Therefore, as confirmed by fastness examination including salt spray, double- 85, light bleaching, and adhesion tests, the laser-printed structural colors on TiN-TiAlN hybrid films are rather durable. In summary, this work paves an appealing approach for high-throughput inkless full-color printing and holds great potential in practical application. I recommend publication of the manuscript after the following minor revisions.

1. In section 3, the authors proposed the utilization of circular polarization to avoid the formation of laser-induced nanoripples, but they did not show the experimental results. The authors should provide more details in the mechanisms of laser-induced nanoripples, at least in the supplementary materials, and explain clearly why the circular polarization is required. Moreover, a comparison between linear- and circular polarization-induced structural colors is required.

2. In Figure 3(a-d), a SEM image and corresponding EDX maps of a representative laser-written area are shown. The EDX maps confirm laser-induced oxidation. However, as demonstrated by the authors, in addition to oxidation-caused structural colors, some colors are formed via ablation. Therefore, the authors should also show EDX maps of some ablative areas.

3. In the manuscript, the depth of TiN reflector is 50 nm. TiN thin film will be semitransparent if its thickness is less than 50 nm. Therefore, if the TiN layer is thinner, can the Si substrate influence the structural colors?

4. It is necessary to provide the refractive indices of TiN and TiAlN to show the material loss for the completeness of data.

5. Relevant work on structural colors with FP nanocavities should be referred for comparison to demonstrate the specific advantages of the approach in this work, e.g. Yang et al. *Advanced Optical Materials*, 4 (8), 2016, 1196; 5 (10), 2017, 170029.

Dear Reviewers,

Thanks very much for taking the time to carefully read our manuscript and for the valuable comments you have provided. These comments helped us in improving the quality of our manuscript. We agree with all your comments, and we corrected point by point the manuscript accordingly.

Please find below our detailed response to each of the comments, and thanks again for your consideration of our manuscript.

Best regards,

Min Qiu

Reviewer #1

Comment summary: The authors report the high-speed laser colouring of a multilayer thin film system consisting of a TiAlN layer on top of a TiN layer. The colours are reported to come from laser induced oxidation of the TiAlN layer which results in reflection of incoming visible light at the oxide-TiAlN and TiAlN-TiN interfaces.

The novelty lies in the combination of techniques where laser-induced oxidation (formation of the alumina layer in other paper - see below - was formed using atomic layer deposition or electrodeposition) is used to tune the oxide thickness on top of the TiAlN-TiN system, therefore tuning the resulting colours. In addition, the colours are automatically protected by the alumina coating, as demonstrated by the authors via various durability tests.

Reply: We thank the reviewer for careful studying our manuscript and pointing out the novelty of our work.

Comment 1: The creation of angle independent colours from multilayered systems is however not new. The authors recently published another article showing angle-independent colour formation on the same material system: J. Geng, L. Shi, J. Ni, Q. Jia, W. Yan, and M. Qiu, "Wear-resistant surface coloring by ultrathin optical coatings," PhotoniX, vol. 3, no. 1, p. 14, Jun. 2022, doi: 10.1186/s43074-022-00061-5.

Other authors have also used multilayered systems to create angle independent colours:

C. Yang et al., "Compact Multilayer Film Structure for Angle Insensitive Color Filtering," Sci Rep, vol. 5, no. 1, Art. no. 1, Mar. 2015, doi: 10.1038/srep09285.);

K. Feng et al., "All-dielectric thin films based on single silicon materials for angle-insensitive structural colors," Opt. Lett., OL, vol. 46, no. 20, pp. 5161–5164, Oct. 2021, doi:10.1364/OL.441534;

Z. Li, S. Butun, and K. Aydin, "Large-Area, Lithography-Free Super Absorbers and Color Filters at Visible Frequencies Using Ultrathin Metallic Films," ACS Photonics, vol. 2, no. 2, pp. 183–188, Feb. 2015, doi: 10.1021/ph500410u.

Reply: Thanks for suggesting these interesting articles, which have been cited in the revised manuscript. Indeed, in recent years, increasing research efforts have been put into the viewing angle-independent structural colors, because this is one of the key performances for practical applications of structural colors. However, in the literature as suggested by the reviewer (Yang et al. *Sci. Rep.* 5, 09285, 2015; Feng et al. *Opt. Lett.* 46, 5161 2021; Li. et al. *ACS Photonics* 2, 183, 2015), the multilayer films behave more as a single-color filter rather than high-resolution and colorful patterns. Actually, it is challenging to produce large-scale colorful patterns on these kinds of films, because their colors are dominantly determined by the dielectric spacer layer (Yang et al. in *Advanced Optical Materials*, 4, 2016, 1196 and *Advanced Optical Materials* 5, 2017, 1700029), which is incompatible with the top-down techniques such as laser fabrication.

In our previous work, we demonstrated another viewing angle-independent colors that consist of only two layers, that is, lossy dielectric (TiAlN) on metallic (TiN) coating. In this scenario, the colors are dependent on the topmost dielectric layer, which is expected to be compatible with laser-based surface processing. Therefore, in this manuscript, we demonstrate in detail the combination of ultrafast laser nanofabrication and TiAlN-TiN hybrid films to simultaneously achieve large-scale, high-throughput, wide-gamut, long-stability, and angle-insensitive patterning.

In other words, our previous work provides a drawing canvas, while this work achieves vivid painting on the canvas, which has several important advantages. Firstly, compared to the conventional nanofabrication techniques such as electron beam lithography or focused ion beam milling, our laser-based technique provides a better brush for high-speed (cm^2/s) and large-scale (4-inch) painting. More importantly, we demonstrate that lasers can paint on surface with high roughness, which is challenging for other large-scale nanofabrication techniques such as nanoimprinting. Secondly, compared to the bulky substrates or multilayer thin films such as FP-cavities, our TiAlN-TiN bilayer film offers a better canvas, which is incompatible with the top-down nanofabrication techniques.

Finally, we would like to point out that in the literature, efforts are mainly put into improving one or two performances of the structural colors, such as the viewing angle-independent or wide gamut. However, in this manuscript, we achieve an excellent overall comprehensive performance, which facilitates a big step towards the practical applications of structural colors. Furthermore, from the perspective of physics, we find that laser-induced oxidation on TiAlN does not only change the colors, but also significantly broadens the pristine gamut.

Comment 2: The creation of structural colours using oxidation is not new either,

especially the addition of an alumina layer to create/tune colours with a similar gamut: X. Wang, D. Zhang, H. Zhang, Y. Ma, and J. Z. Jiang, "Tuning color by pore depth of metal-coated porous alumina," Nanotechnology, vol. 22, no. 30, p. 305306, Jul. 2011, doi: 10.1088/0957-4484/22/30/305306.

J.-M. Guay et al., "Laser-written colours on silver: optical effect of alumina coating," Nanophotonics, vol. 8, no. 5, pp. 807–822, May 2019, doi: 10.1515/nanoph-2018-0202.

Reply: Thanks again for sharing these interesting papers. Our laser-induced formation of alumina layer for coloring and *in-situ* protection has advantages with respect to that demonstrated in these references.

First, in the article (J.-M. Guay et al., "Laser-written colours on silver: optical effect of alumina coating," Nanophotonics, vol. 8, no. 5, pp. 807–822, May 2019, doi: 10.1515/nanoph-2018-0202.), the post-coating of alumina by atomic layer deposition is used to protect the plasmonic nanoparticles. But unfortunately, the *ex-situ* deposition of alumina on the plasmonic nanoparticles will change the initially generated colors, which requires complex compensation to recover the colors, and thus is not convenient for practical applications. Further, the generated gamut is still relatively narrow.

Second, in article (X. Wang et al., "Tuning color by pore depth of metal-coated porous alumina," Nanotechnology, vol. 22, no. 30, p. 305306, Jul. 2011, doi: 10.1088/0957-4484/22/30/305306.), the alumina is used as transparent spacer layer of FP-cavities rather than as protection layer. Further, it needs complicated processes to get colourful patterns. For instances, as a first step, Al foils have to be annealed at 500°C under vacuum ambient for 5 hours to remove the mechanical stress, and then electrochemically polished by a mixture of HClO₄ and C₂H₅OH (volume ratio 1:9); in second step, it needs anodization lasted for 2 hours under an electrolyte temperature of 3°C; Next, the samples are immersed in a mixture of 6.0 weight percent (wt%) H₃PO₄ and 1.8 wt% CrO₃ at 60°C for 4 hours to remove the oxide layer. Finally, for colourful patterning, a positive or negative mask with expected pattern is required.

In summary, our top-down technique holds advantages in wide-gamut coloring and *in-situ* protection, process simplicity (single-step direct laser write), and in that no masks, stamping die or chemical processing are required.

Comment 3: While the rendered colours are mostly a result of the complex interference effects, there may be some structural effects are well that should be discussed. It is possible to see a little bit of that surface morphology in Fig. 3(f-j) and the AFM image Fig. 3(k-l) but only for 1 condition. Larger area SEM images or AFM images should be provided for the various conditions shown in Fig. 3(f-j). If surface morphology doesn't affect the Hue, it might at least affect the brightness and thus the resulting colour. This should be discussed in the paper especially when the roughness of larger than the layer thickness.

Reply: as suggested by the reviewer, we have provided high-resolution SEM images and AFM images for the various conditions in Fig. 3(f-j). Please note that Fig.3 in the

initial submission has been changed to Fig. 4(f-j) in the revised manuscript. We have also discussed the influence of surface roughness in the reflected colors by FDTD-based numerical simulations.

As shown in Fig. R1, the light-blue and green colors have surface cracks that may be attributed to temperature gradient and thermal stress. Especially, the green color swatch exhibits very low roughness. For blue, red and yellow color swatches, which require either higher laser power or more accumulated pulses, exhibit ablative debris on oxide layer and thus having a higher surface roughness, as confirmed by the AFM images. Indeed, these surface structural effects may influence the colors.

Figure R1. (a-e), SEM image and corresponding EDX maps of a laser-written area with green color. Cross-sectional view images that are obtained by focused ion beam milling [(f-i)-(j-i)], high-resolution SEM images [(f-ii)-(j-ii)] and AFM images [(f-iii)-(j-iii)] of five different laser-coloured areas that are obtained by different scanning speeds or pulse energies. The insets show their corresponding color swatches.

In order to study the influence of surface roughness in the reflected colors, we first investigate the material components of the deposited nanoparticles by high-resolution EDX maps, as shown in Fig. R2. We find that the dominant component of the redeposited ablative nanoparticles is aluminium oxide.

Figure R2. High-magnification EDX maps of five different color palettes, the same as that displayed in Fig. R1.

Next, we study the influence of surface roughness in reflectance spectra by FDTD-based numerical simulation. We place periodic alumina nanoparticles with different diameters while a constant period on the surface of our hybrid film to represent the different roughness, as shown in the inset of Fig. R3(a). More details of the numerical simulation can be found in Supplementary Section 6.

As shown in Fig. R3(a), unlike the metallic nanoparticles that generally exist plasmonic resonances, the alumina dielectric nanoparticles have no additional resonance absorption and do not change the reflectance spectral profile of the hybrid films. The reflectance dips at ~ 650 nm which originate from the non-trivial phase shift at TiN-TiAlN interface are also insensitive to the aluminium nanoparticles. However, reflectance dips at ~ 300 nm that is related to Al_2O_3 -TiAlN interface slightly redshift (as indicated by the solid arrow in Fig. R3a) with the increase of particles' diameter. This may be attributed to that the bigger nanoparticles have higher effective thickness of oxide layer.

Nevertheless, the corresponding variations of colors in CIE1931 diagram is fairly small, as shown in Fig. R3(b). From the AFM images we also find that most of palettes that are obtained at different pulse energies and scanning speeds have comparable roughness while completely different colors. Therefore, we believe that the colors are from thin-film interference rather than the random surface nanoparticles.

Figure R3. (a) simulated reflectance spectra of hybrid films when existing different diameters of alumina nanoparticles. Inset: scheme of the geometry for numerical simulation. (b) Numerically simulated CIE 1931 color coordinates for different diameters of alumina nanoparticles on the surface of a Al₂O₃-TiAlN-TiN hybrid film.

But indeed, as mentioned by the reviewer, the roughness will at least affect the brightness. This is also confirmed by the simulations, as shown in Fig. R3a, the reflectance decreases with the increase of alumina nanoparticles' diameter.

Comment 4: The authors claim a resolution of 25000dpi based on a spot size of 1 μ m. However, In Fig. S1, the area has a diameter closer to 2 or 3 microns. Why not calculate the dpi based on the actual spot size? Furthermore, it would be good to demonstrate laser printing using this spot size on a larger area (like in Fig.2 a) for instance.

Reply: We are sorry that using the optical microscope to measure the laser spot approaching to diffraction limit has large error. Therefore, we employed an electron microscope to precisely measure the size of a single spot and single line. Figure R4 show the SEM images of laser-coloured surfaces that are produced by a single shot and line scanning, respectively. The sizes of pixels are measured to be 1-1.5 μ m, corresponding to printing resolution of 17000-25000 dpi. Therefore, in the revised manuscript, we rewrote the sentences: “The resolution can be improved by utilizing a smaller laser beam spot. For instance, when focusing the laser beam by an objective lens (NA = 0.9), the printing resolution exceeds 10000 dpi (Fig. S3c-S3e). The spot sizes of pigments are generally on the order of 25 μ m, resulting in a resolution of ~1000 dpi. Therefore, the laser-printing ink-free structural colors can have one order of magnitude higher resolution than the conventional printers. The SEM images in Fig. 3(d, e) show nanoparticles depositing on cracked surface, in agreement with that are produced by loosely focused beam.”

Figure R4. Optical microscopy images and corresponding SEM images that are generated by line scanning (c, d) and single spot (e) when the laser beam is tightly focused by an objective lens with NA = 0.9. (f) optical microscopy images of color swatches (Tetris game) that are produced by meander scanning.

However, we would like to clarify that, for high-resolution coloring, the laser beam was coupled into an optical microscope and focused onto samples by an objective lens with NA = 0.9. The working distance and laser Rayleigh length are very short, but we do not have an automatic real-time focus control system for laser processing. Therefore, the system is highly sensitive to mechanical vibration, surface relief and so on. As a result, the uniformity of colors over large areas is not excellent, as shown in Fig. R4f. Nevertheless, the resolution is not the most critical performance for many practical applications, because a printing resolution of 360 dpi when using loosely focusing is generally acceptable.

Comment 5: In Fig. 2(d), the conditions (parameters) used to create the curves should be written in the Figure caption (e.g. pulse energy, scanning speed, etc.)

Reply: the experimental parameters including the pulse energy and scanning speed used to create the curves have been added in Figure 2.

Comment 6: Regarding the angle-independence of the colours, the authors should render the colours from the spectra in Fig. 2f to really prove that the colours do not change as small changes in the spectra can lead to observable changes in the rendered colours. The authors show the Hue vs angle to prove angle independence, but the colour appearance is not only provided by the Hue but also by the brightness, etc. Why not show the colours obtained experimentally as a function of viewing angle?

Reply: thanks for this suggestion. In the revised manuscript, we have taken photographs of the laser-induced colors at different viewing angles and confirmed that the colors are indeed insensitive to the viewing angles, as shown in Fig. R5.

Figure R5. Photographs of a laser-written color swatch taken at five titled angles.

Regarding the reflectance spectra at different angles, we would like to clarify that a collimated light beam incident from a specific angle was used for these measurements, therefore, the reflection intensity varies with the incident angles. However, in a real environment, the ambient light illuminates the samples from a wide range of angles. As a result, the observed brightness does not obviously change with viewing angles.

Therefore, in this case the Hue values, that are mainly determined by the peak of the reflection spectra, are more important. From the spectra in Fig. R6, we find that the peaks of the reflection spectra and thus the Hue values do not shift at titled angles. We have measured many colors and confirm that the peaks of the reflection spectra do not change with angles.

Figure R6. Reflectance spectra versus incident angles of a representative color swatch.

In order to demonstrate our statement in a better way, in the revised manuscript, the photographs of a color swatch (Fig. R5) have been added into Fig. 2 in the main text, and the reflectance spectra at different angles have been moved to Supplementary Materials.

Comment 7: The authors say “Therefore, compared to laser-written colors directly on metals, [40] our technique exhibits the advantages of viewing angle-independence and wider gamut.” suggesting that laser-written colours are not viewing angle independent. However, there are many reports in literature of angle-independent colours based on surface plasmons. (e.g. ref 5 and 41 in the authors list).

Reply: we are sorry that our statement here is unclear. Indeed, laser-induced plasmonic colors on noble metals are also viewing angle-independent, but we would like to mention the interfering colors that come from laser-induced transparent oxide layer on metals are sensitive to the viewing angles that have been widely reported in the literature. Actually, in our lab, we have also performed laser coloring on metals via oxidation layer and found that, indeed, the colors are sensitive to the viewing angles (similar as soap bubbles). The reason is that when changing the viewing angles, the accumulated phases by the transparent oxide layer are varied. However, in our new scheme that is based on lossy dielectric, the strong non-trivial phase shifts at the interfaces reduce the influence of the thickness in the phase changes.

We rewrote this sentence to be: “**Therefore, compared to structural colors from laser-induced oxide layer on metals, our coloring on lossy dielectric exhibits the advantages of viewing angle-independence.**”

Comment 8: When describing Figs. 4a and b, the deposition of the 50-nm-TiN and 50-nm-TiAlN layers on the Si substrates should be mentioned as it is not clear from the text only (even though it is mentioned in the figure caption).

Reply: in the revised manuscript text, we added a sentence: “TiN film with a thickness of 50 nm and TiAlN film with a thickness of 60 nm are subsequently deposited onto the substrates by RF magnetron reactive sputtering.”

Comment 9: Fig. 1 – “a)” is missing in the Figure caption. In the same figure caption, the comment “Photograph of surface coloring with various TiAlN and AlN coatings on $t_3=50$ nm TiN. The substrates are Si wafers or glasses.” is vague and does not allow to identify where the Si or glass substrate were used. Do they make a difference in the colours observed? Why two different substrates were used? This is the only location where glass is mentioned in the text.

Reply: We have added the Figure caption for Fig. 1(a). Please note that in the revised manuscript, Fig. 1(a), i.e., the experimental scheme, has been individually displayed as Figure 1, and the other sub-figures in Fig. (1) have been moved to Figure 2.

Figure R7. (Figure 1 in the manuscript) Scheme of surface coloring by ultrafast laser. The structural colors originate from laser-induced oxide layer on highly absorbing while extremely hard ceramic TiAlN film. The TiN film behaves as a reflector. QWP: quarter-wave plate. Large-scale printing is achieved by meander scanning. The interference of light that transmits and reflects at the interfaces gives rise to strongly resonant absorption and displays tunable colors. t_1 : thickness of laser-induced oxide layer (medium 1); t_2 : thickness of remained TiAlN film (medium 2); t_3 : thickness of TiN (medium 3). Laser simultaneously changes t_1 and t_2 , which result in various reflected colors (inset).

Regarding the substrates that are Si wafers or glasses, when the TiN is thick enough (>50 nm) such as in our experiments, we find that the substrates play negligible roles. However, in order to keep consistency in the whole manuscript, we have taken another photo that the substrates are all Si wafers, as shown in Fig. R8.

Figure R8. Photograph of surface coloring with various thickness of AlN (t_1) subsequent with TiAlN ($t_2=0, 10, 15, 25, 35, 50$ in unit of nm) coatings on 50 nm TiN. The substrates are 2-inch Si wafers. On each sample, three different thickness of AlN are employed: top $t_1 = 0$; middle $t_1 = 25$ nm; bottom $t_1 = 50$ nm. Please note that masks were used to change the thickness of AlN in different areas during film coatings.

Finally, the paper should be proofread to correct the remaining mistakes, discourse errors and writing inconsistencies.

Reply: thanks again for careful reading our manuscript and comments, which are very helpful to improve the quality of this paper. We have carefully revised the manuscript according to your comments, corrected the remaining mistakes and writing inconsistencies.

Reviewer #2

Comment summary: The manuscript presents laser coloring method to create optical absorbers on TiAlN & TiN hybrid film, for full-color inkless printing use. Comparing to the conventional nanofabrication techniques, the laser method had unique advantages in the cost and growth time, which is very important for practical applications. I found this idea interesting.

Reply: Thanks very much for careful reading our manuscript, pointing out the advantages of our technique and providing valuable comments that allow us to greatly improve the quality of the manuscript. We agree with all your comments, and we corrected point by point the manuscript accordingly.

Comment 1. In introduction, the authors should properly review the previous work in the field and detail what the deficiencies you are targeting at, followed by how their new laser method is working.

Reply: thanks for your suggestions. We have accordingly revised the introduction

section. In the first paragraph, we start to introduce the general advantages of structural colors with respect to the conventional dyes and pigments-based colors, and then point out that it is challenging to produce large-scale colourful patterns by the conventional nanofabrication techniques. The key sentences are: “the development of structural colors-based ink-free printers is demanded on looming.

...However, surface coloring in terms of the conventional nanofabrication techniques, such as electron beam lithography, [37, 38] focused ion beam milling [39] and nanoimprinting lithography, [40] is facing the nanoscale and macroscale processing barrier....”

In the second paragraph, we first point out that lasers are capable of large-scale surface coloring with high throughput and enable to fabricate on rough surface that are challenging for the conventional technique such as electron beam lithography and nanoimprinting. Then, we briefly introduce three state-of-the-art laser coloring techniques. They are “plasmonic colors from randomly self-organized metallic nanoparticles, diffractive colors from laser-induced periodic surface structures (LIPSS, i.e., nanogratings), and interfering colors from thin films including transparent oxide layer and FP-cavities.” The key sentences are: “As an alternative, surface coloring by lasers, exhibiting high throughput of $>10 \text{ mm}^2/\text{s}$, [41] is appealing to overcome this barrier.

...Laser coloring generally contains three approaches that are based on different mechanisms:...”

In the third paragraph, we summary the limitations of laser-induced plasmonic colors and LIPSS, followed by the limitations of laser-induced oxide layer on metals in the fourth paragraph and limitations of the FP-cavities in the fifth paragraph. The key sentences are: “The laser-produced plasmonic colors on metals exhibit advantages of viewing angle-independent but face the problems of narrow gamut ($\sim 15\% \text{sRGB}$) [42] and low stability. LIPSS exhibit iridescence that strongly limits its practical applications in patterning.

Another widely adopted laser coloring technique is in terms of laser-induced oxide layer, generally occurring on metallic surfaces such as titanium alloy or stainless steel. [41, 45, 49] This approach exhibits the advantages of high stability and productivity. However, its gamut is also narrow ($\sim 35\% \text{sRGB}$). Furthermore, the colors from laser-induced oxide layer are dependent on viewing angles

The structural colors originating from FP-cavities can be insensitive to the viewing angles However, it is challenging to directly modify the spacer layer by lasers because of the high reflection of the metallic film this technique is also facing the problem of narrow gamut ($\sim 25\% \text{sRGB}$). Further, both plasmonic colors and FP-cavities rely on noble metals such as Au, Ag and Cu, exhibiting low wear resistance and thus poor abrasion stability. ”

Next, we point out that structural colors with wide gamut, durable, large-scale and viewing angle-insensitive are required for practical applications, and followed by the

mechanisms of our new idea, that is, the use of TiAlN-TiN hybrid film as 'inorganic ink' to subtly achieve an all-in-one solution. Below, we detailly introduce why our lossy dielectric-on-metals have angle-insensitive colors (non-trivial phase shift), wide gamut (double resonance), fast printing speed (thin film instead of complex nanostructures) and long stability (self-formed alumina).

Comment 2. The scheme in Figure 1a is not very clear. The label of Figure 1a is missed in figure caption. The five rectangles in Fig. 1a are not easy to understand. The colors represent medium 1,2 &3 need to be consistent in the picture to avoid possible misunderstandings.

Reply: Figure 1a has been redrawn, as shown in Fig. R7. Hopefully, the new version is clear for the readers. Further, the colors representing medium 1, 2 & 3 has been consistent in Figure 1 and Figure 2 to avoid possible misunderstandings.

Comment 3. The results on the printing speed of 1.4 cm²/s and resolution of 25000 dpi are based on the speculations but not on the results of the real printing.

Reply: in order to test the actual printing speed, we performed experiments at 10 m/s scanning speed at resolution of 250 dpi, please refer to the results in Supplementary Movie 1. The following Figure R9 shows the coloured surface with a size of 10 cm² that was printed in 1 second (10 cm²/s).

Figure R9. Printing a 3.2cm×3.2 cm square in 1 second.

Regarding the highest printing resolution, we employed an electron microscope to precisely measure the size of a single spot and single line. Figure R10 show the SEM images of laser-coloured surfaces that are produced by a single shot and line scanning, respectively. The sizes of single units are 1-1.5 μm , corresponding to printing resolution of 17000-25000 dpi. Therefore, in the revised manuscript, we rewrote the sentences: "The resolution can be improved by utilizing a smaller laser beam spot. For instance, when focusing the laser beam by an objective lens ($\text{NA} = 0.9$), the printing resolution exceeds 10000 dpi (Fig. S3c-S3e). The spot sizes of pigments are generally on the order of 25 μm , resulting in a resolution of ~ 1000 dpi. Therefore, the laser-printing ink-free structural colors can have one order of magnitude higher resolution than the conventional printers. The SEM images in Fig. 3(d, e) show nanoparticles depositing on cracked surface, in agreement with that are produced by loosely focused beam."

Figure R10. Optical microscopy images and corresponding SEM images that are generated by line scanning (c, d) and single spot (e) when the laser beam is tightly focused by an objective lens with NA = 0.9. (f) optical microscopy images of color swatches (Tetris game) that are produced by meander scanning.

However, we would like to clarify that, for high-resolution coloring, the laser beam was coupled into an optical microscope and focused onto samples by an objective lens with NA = 0.9. The working distance and laser Rayleigh length are very short, but we do not have an automatic real-time focus control system for laser processing. Therefore, the system is highly sensitive to mechanical vibration, surface relief and so on. As a result, the uniformity of colors over large areas is not excellent, as shown in Fig. R10f. Nevertheless, the resolution is not the most critical performance for many practical applications, because a printing resolution of 360 dpi when using loosely focusing is generally acceptable.

4. The author said “Plotting the reflectance spectra of the matrix palette into CIE 1931 chromaticity diagram, as shown in Fig. 2(e), we verify that the laser-printed colors have a wide gamut (~90% sRGB)”. But how did they come out with the result of “~90% sRGB”? Because the area of the circles in Figure 2e seems to be much smaller than the triangle representing sRGB area. Besides, the “~90% sRGB” is not a excellent performance in the wide gamut in optical meta-surface field, considering many previous works has reached larger gamut than the sRGB area.

Reply: the gamut is estimated by using an ellipse to surround all the measured points in the CIE diagram. The area of the ellipse is ~90% of the area of sRGB, that is, the triangle representing sRGB. Indeed, as mentioned by the reviewer, our laser-written gamut is narrower than many previously reported optical metasurfaces that are fabricated by the conventional nanofabrication techniques such as electron beam lithography or focused ion beam milling.

However, compared to state-of-the-art laser coloring techniques (Fig. R12), our method exhibits a much wider gamut. More importantly, the green colors are difficult to be obtained by these conventional methods. This problem has been solved in our method. By taking advantages of double-resonance occurring at the oxide/TiAlN interface and at TiAlN/TiN interface, we obtain vivid green colors. The plasmonic colors arise from laser-induced randomly self-organized metallic nanoparticles, whose gamut is ~15% sRGB (Fig. R12a). The structural colors from laser-induced oxide layer on metallic surfaces share the same effect of colors from soap bubbles. However, in such cases, the reflected colors are dependent on viewing angles, and its gamut is ~35% sRGB (Fig. R12b). The laser-induced F-P cavities via polymer-assisted photochemical metal deposition (Fig. R12c) have viewing angles-insensitive colors, but the gamut is also rather narrow. **Further, it should be pointed that in the case of laser-induced oxide layer on metallic surface and laser-induced F-P cavities, only one parameter (the thickness of the oxide layer) can be changed, thus its gamut in CIE diagram follows a specific curve, colors inside the curve cannot be directly obtained.** However, in our method, **we simultaneously change the thickness of the oxide layer and the remaining lossy-dielectric layer (two parameters)**, therefore, the colors are more diverse (**colors inside the curve can also be directly obtained**) and obviously wider than the initial gamut of the pristine films (stars in Fig. R11). Therefore, our research highlight is that we simultaneously achieved a very high manufacturing speed and an acceptable gamut.

Figure R11. Measured CIE chromaticity diagram in our experiments. The laser-coloring TiAlN-TiN hybrid films (blue circles) have obviously wider gamut than the pristine films (red stars).

Figure R12. gamut of laser-induced plasmonic colors on noble metals (a), laser-induced oxide layer on titanium alloy (b), and laser-induced FP-cavities (c). The numbers in (c) correspond to the thickness of the topmost Ag film. Please note that these methods are difficult to generate green colors.

Comment 5. The data appeared in Figure 2e should be specified clearly if they came from simulation or experiment? Fig. 2g is not seriously based on data and references. The comparison is not scientifically sound and thus it is not suggested to put here.

Reply: We have clarified that the data appeared in Figure 2e is obtained from experimental results. We measured the reflection spectra of different laser-produced colors and plotted them into the CIE 1931 chromaticity coordinate.

Regrading to Fig. 2g, we have revised it based on reference (APL Photonics 4, 051101, 2019). This review article has systematically compared the performances of different state-of-the-art laser coloring techniques, as shown in Table R1.

Technique	Surface oxidation	LIPSS	Nanostructures/particles
Stability	High	Moderate	Low
Operation cost	Low	High	High
Productivity	High (0.5–60 mm ² /s)	Low (0.005–0.8 mm ² /s)	Moderate (0.1–36 mm ² /s)
Resolution (focused beam spot) (μm)	13–350	5–300	14–50

Table R1. A comparison of three state-of-the-art laser coloration techniques (APL Photonics 4, 051101, 2019) in terms of their performances on stability, operation cost, productivity, and

spatial resolution.

In addition, we add another two indices according to the Technology-performance indicators for colour technologies in literature (Nature Reviews Materials 2, 16088 (2017)) and compare our result with the commonly used laser-coloring techniques, as shown in Fig. R13. It indicates that our technique exhibits a high overall performance and thus providing an all-in-one solution for practical applications. For instance, operating at comparable laser repetition rate, our productivity reaches to $10 \text{ cm}^2/\text{s} = 1000 \text{ mm}^2/\text{s}$, which is at least one order of magnitude higher than that in Table R1.

Figure R13. technology-performance indicators expressing on low, moderate and high level that provides an overview of the state-of-the-art laser coloring techniques.

Comment 6. The authors should show the laser-written area in SEM images with different areas and viewing scale to see more units, other than only a very limited area as shown in Fig. 3 and Fig. S1. An overview and clear images showing the quality of the structure created by the laser method is very important for structure color. Continuous or discrete unit structures with small area or large areas, should be created using different manufacturing parameters. I would like to see some structures in SEM images in the large-scale laser color printing examples in Fig. 4 and Fig. Fig. S3.

Reply: as suggested by the reviewer, we have performed extensive surface analyses on more coloring units by using SEM and EDX. Figure R14 show the two-dimensional EDX maps of several representative color palettes. The comparison between laser-modified areas (left-bottom areas in the SEM images) and the pristine films confirm that all generated colors are related to laser-induced oxidation, that is, nitrogen has been partially replaced by oxygen. With regard to blue, red and yellow colors that occur at high totally accumulated laser fluence, we find that the EDX signal of aluminum is reduced. This indicates that a part of aluminum has been ablated at high accumulated

fluence. As a result, the EDX intensity of silicon substrate is enhanced within the laser-irradiated areas. Indeed, the ablation occurs at high accumulated fluence was also confirmed by the FIB-based cross-sectional view, as discussed in the manuscript (Figure 4f-4j). These surface analyses are also in agreement with the numerical predictions in CIE diagram (Fig. 2b), which suggests that the green-color has thickest TiAlN while the yellow-color has thinnest TiAlN.

Figure R14. Low-magnification SEM images and EDX maps of five different color swatches. The laser modified areas locate in the left-bottom regions in the SEM images.

Next, we also obtained high-resolution SEM and AFM images of different color palettes, as shown in Fig. R15. The light-blue and green colors have surface cracks that may be attributed to temperature gradient and thermal stress. Especially, the green color swatch exhibits very low roughness. For blue, red and yellow color swatches, which require either higher laser power or more accumulated pulses, exhibit ablative debris on oxide layer and thus having a higher surface roughness, as confirmed by the AFM images.

We further investigate the material components of the laser-induced rough surface by high-resolution EDX maps, as shown in Fig. R16. We find that the dominant

component of the redeposited ablative nanoparticles is aluminium oxide. It should be pointed out that most of palettes that are obtained at different pulse energies and scanning speeds have comparable roughness while completely different colors. Therefore, we believe that the colors are from thin-film interference rather than the random surface nanoparticles, because these small dielectric alumina nanoparticles do not cause additional resonant absorption. Nevertheless, the scattering of surface roughness may influence the brightness of the reflected colors.

Figure R15. Cross-sectional view images that are obtained by focused ion beam milling [(f-i)-(j-i)], high-resolution SEM images [(f-ii)-(j-ii)] and AFM images [(f-iii)-(j-iii)] of five different laser-coloured areas that are obtained by different scanning speeds or pulse energies. The insets show their corresponding color swatches.

Figure R16. High-magnification EDX maps of five different color palettes.

Regarding the structures in SEM images in the large-scale laser color printing examples in Fig. 4, we have shown them in the manuscript (Fig. 5a-5c). However, for the large-scale image in Fig. S3, because it was produced on unpolished surface, the roughness is too high for SEM imaging.

Comment 7. The durability test on the laser colors was shown in Fig. S4. The photographs on the difference are suggested to see the durability, which was claimed as one of the advantages of this laser method.

Reply: In the revised supplementary materials, we supply the detailed color differences of four coloured areas after various aging tests, as shown in Fig. R17(a),

The color difference $\Delta E_{a,b}^* = \sqrt{(\Delta L)^2 + (\Delta a)^2 + (\Delta b)^2}$ was measured in Lab color space. Here ΔL represents a lightness difference between a pristine and tested sample, Δa denotes the difference in redness or greyness and Δb is blueness-yellowness differences. The corresponding photographs before and after 120 hours of various aging tests are shown in Fig. R17b-R17d.

Figure R17. (a) measured color differences of four coloured areas after various aging tests for 120 hours. (b-d) Photographs of samples that are used as references and after various aging tests.

Comment 8. In Figure 4, the author showed three large-scale laser color printings. However, I had a few questions about this exhibition. First, the author wrote “we find that the generated colors on rough surfaces delivers more uniform brightness than that on polished ones”. But the only thing I can tell from Figure 4a&b is that the colors in Figure 4a had a higher saturation than Figure 4b. If the author is talking about the “more uniform brightness” from different observation angles, he should show the photos in different angles and analyze the results.

Second, the Figure 4c had a blue gradient on the steel foil, in the top of the picture. Did the gradient come from the substrate itself, like the uneven lighting? In a paper about colors, photo should be taken by the camera with accurate settings to show the original colors of the metasurface. The same issue also existed in Figure S3; the uneven flashlight made the 64 identical samples look different from each other. Finally, I suggest the authors to draw a CIE diagram concluding colors in Figure 4a for comparison, showing the real printing quality of this laser method.

Reply: indeed, we would like to talk about “more uniform brightness” of colors on rough surfaces than that on polished surfaces. The reason is that on polished surfaces, the visualized colors are from mirror reflection, while on rough surfaces the observed colors mainly come from diffuse reflection. When the reflected light enters the detector (camera or eye), the brightness of a polished surface is very high, as shown by the upper image in Fig. R18(a). However, at a tilted angle, the areas where the mirror reflection does not enter the detector are obviously dark (upper image in Fig. R18a).

In other words, on polished surface, the observed brightness significantly depends on the angle of mirror reflection. However, when coloring on rough surfaces, the diffuse reflection ensures that the brightness is more uniform at different viewing angles (bottom images in Fig. R18a and Fig. R18b).

Regarding the higher saturation of the polished surface than that of the rough surface, the reasons can be attributed to that--on polished surfaces, the light is totally reflected by the thin-film coatings, while on rough surface, the scattering of white light by the side walls of the surface relief where there are no coatings can reduce the saturation.

Figure R18. photographs of a laser-coloring surface on polished (upper images) and on unpolished surface (bottom images)

Second, regarding the color gradient on the steel foil and on the 64 identical samples in Supplementary Materials, they indeed come from the uneven flashlight. But unfortunately, the samples have been selected as gifts for the first-year undergraduate students at Westlake university, therefore, we cannot take a new photo.

Finally, we would like to explain that the colors of large patterns were manually selected by naked eye for proof-of-principle experiments. Precise coloring has to be done using vision alignment software with targeted pattern recognition, which unfortunately is still not accessible in our lab.

Reviewer #3

Report summary

Geng et al. demonstrated the use of ultrafast lasers to directly write large-area (wafer-scale) and wide-gamut (~90% sRGB) structural colors that are insensitive to viewing

angles. The printing speed reaches to $1.4 \text{ cm}^2/\text{second}$, which is impressive, especially compared to the conventional nanofabrication techniques such as electron beam lithography and focused ion beam milling. Furthermore, it should be pointed that although surface coloring by pulsed lasers-induced oxidation has been previously demonstrated on bulky materials such as stainless steel and silicon, nevertheless, this technique is facing the stubborn problems of narrow gamut and viewing angles-dependent colors. In this work, the authors have solved these problems by utilizing TiN-TiAlN hybrid films as 'inorganic ink'. Another big advantage of such hybrid films is that these materials exhibit extremely high hardness and stability. Therefore, as confirmed by fastness examination including salt spray, double- 85, light bleaching, and adhesion tests, the laser-printed structural colors on TiN-TiAlN hybrid films are rather durable. In summary, this work paves an appealing approach for high-throughput inkless full-color printing and holds great potential in practical application. I recommend publication of the manuscript after the following minor revisions.

Reply summary:

We thank the Reviewer for careful reading our manuscript and recommending the publication. We agree with all your comments, and we corrected point by point the manuscript accordingly.

Comment 1. In section 3, the authors proposed the utilization of circular polarization to avoid the formation of laser-induced nanoripples, but they did not show the experimental results. The authors should provide more details in the mechanisms of laser-induced nanoripples, at least in the supplementary materials, and explain clearly why the circular polarization is required. Moreover, a comparison between linear- and circular polarization-induced structural colors is required.

Reply: Laser-induced nanoripples are also called as laser-induced periodic surface structures (LIPSS), which is a universal effect that can be observed on almost any material after the irradiation by linearly polarized laser beams, particularly when using ultrashort laser pulses with durations in the picosecond to femtosecond range. LIPSS is generally produced by the far-field interference between incident light and scattered surface waves associated with near-field enhancement in vicinity of scatters. As shown by numerical simulations in Fig. R19 (a), when the incident light is linearly polarized, the orientation of near-field enhancement and far-field interfering patterns are perpendicular to incident light polarization (E_0), as a result, nanoripples with orientation perpendicular to laser polarization are observed in experiments, as shown in Fig. R19(c). Instead, when circular polarization is utilized, the field enhancement is isotropic in each direction (Fig. R19b), as a result, the laser-modified surface is more uniform (Fig. R19d).

Figure R19. (a, b) FDTD-based numerical simulation of electric field distribution at TiAlN–air interface in the presence of an alumina nanoparticle with diameter of 300 nm. The light source is a plane wave at 1030 nm with linear (a) and circular (b) polarization. Optical microscope images of laser-modified TiAlN surfaces by linear (c) and circular (d) polarization while at the same laser fluences and scanning speeds. The thickness of TiAlN and TiN are both 50 nm in simulations and experiments.

The light diffraction by the LIPSS-induced nanograting will generate iridescent colors, as shown in Fig. R20.

Figure R20. structural colors from LIPSS, originating from grating-induced light diffraction.

Comment 2. In Figure 3(a-d), a SEM image and corresponding EDX maps of a representative laser-written area are shown. The EDX maps confirm laser-induced oxidation. However, as demonstrated by the authors, in addition to oxidation-caused structural colors, some colors are formed via ablation. Therefore, the authors should also show EDX maps of some ablative areas.

Reply: as suggested by the reviewer, EDX maps on several representative color palettes are performed, as shown in Fig. R21. The comparison of EDX between laser-modified and the pristine areas confirm that all generated colors are related to laser-induced oxidation, that is, nitrogen has been partially replaced by oxygen. With regard to blue, red and yellow colors that occur at high totally accumulated laser fluence, we find that the EDX signal of aluminum is reduced. This indicates that a part of aluminum has been ablated at high accumulated fluence. As a result, the EDX intensity of silicon substrate is enhanced within the laser-irradiated areas. Indeed, the ablation occurs at high accumulated fluence was also confirmed by the FIB-based cross-sectional view, as discussed in the manuscript (Figure 4f-4j).

Figure R21. Low-magnification SEM images and EDX maps of five different color swatches. The laser modified areas locate in the left-bottom regions in the SEM images.

Comment 3. In the manuscript, the depth of TiN reflector is 50 nm. TiN thin film will be semitransparent if its thickness is less than 50 nm. Therefore, if the TiN layer is thinner, can the Si substrate influence the structural colors?

Reply: Indeed, as suggested by the reviewer, the structural colors are also related to the thickness of the TiN which acts as reflector. As shown in Fig. R22(a), when the TiN film is thinner than 50 nm, the reflected colors are highly sensitive to its thickness. As a result, the structural colors can also be influenced by the substrate materials, as compared in Fig. R22(b).

Figure R22. (a) Numerically simulated CIE 1931 color coordinates for several different thickness of TiN layer in unit of nm. The substrate is c-Si wafer, and the thickness of TiAlN is 50 nm. (b) Numerically simulated CIE 1931 color coordinates for two different substrates. The thickness of each thin layer is identical on two different substrates.

Further, we have coated 50 nm of TiAlN on different thickness of TiN, as shown in Fig. R23, the observed colors are in agreement with the numerical simulation in Fig. R22(a). When the thickness of TiN is less than 50 nm, the reflected colors are obviously different.

Figure R23. Photograph of 50-nm-TiAlN coating on different thickness of TiN. The substrates are all 2-inch Si wafers.

Comment 4. It is necessary to provide the refractive indices of TiN and TiAlN to show the material loss for the completeness of data.

Reply: as suggested by the reviewer, we have measured the refractive indices of TiN and TiAlN by a variable angle spectroscopic ellipsometer (Woollam), as shown in Fig. R24. We confirm that TiAlN manifests as a highly lossy dielectric and TiN film behaves as a metallic material. The details are given in Supplementary section 1.

Figure R24. Experimentally measured dielectric constants of TiN and TiAlN that are deposited by RF magnetron reactive sputtering.

Comment 5. Relevant work on structural colors with FP nanocavities should be referred for comparison to demonstrate the specific advantages of the approach in this work, e.g. Yang et al. Advanced Optical Materials, 4 (8), 2016, 1196; 5 (10), 2017, 1700029.

Reply: thanks for sharing these interesting articles. Indeed, the F-P nanocavity is one of the main categories in terms of the mechanism of reflective color filters, as demonstrated by Yang et al. in *Advanced Optical Materials*, 4 (8), 2016, 1196 and *Advanced Optical Materials* 5 (10), 2017, 1700029. The F-P cavities-based reflective color filters generally consist of two metallic layers that are separated by a lossless dielectric layer. The thickness of the lossless dielectric layer determines the reflected colors. The F-P cavities also exhibit the advantages of viewing angles-insensitive structural colors and wide gamut. However, the F-P cavities are incompatible with the laser fabrication techniques, because the topmost metallic layer will inhibit the penetration of laser into the spacer layer. Choi recently demonstrated the use of laser-induced polymer-assisted photochemical metal deposition (PPD) to change the thickness of the topmost metallic layer (*Light: Science & Applications* 11, 84 2022). However, in this case, the gamut is rather narrow.

As an alternative, in our method, the colors are from the resonant absorption of lossy dielectric-on-metal films. The colors are strongly related to the thickness of the topmost layer thickness. Therefore, it is easy to utilize laser to modify the dielectric layer and thus changing the corresponding colors. In the introduction section, we have cited Ref. Yang et al. *Advanced Optical Materials*, 4 (8), 2016, 1196 and 5 (10), 2017, 1700029. And discussed the differences of our work with respect to the F-P cavities-based structural colors.

Reviewer #2 (Remarks to the Author):

The revised manuscript has adequately addressed the raised concerns and is acceptable now.

Reviewer #3 (Remarks to the Author):

The authors well addressed my concerns and I have no more comments.

Reviewer #2

The revised manuscript has adequately addressed the raised concerns and is acceptable now.

Reply: We thank the reviewer for recommending the acceptance of our manuscript.

Reviewer #3

The authors well addressed my concerns and I have no more comments.

Reply: We thank the reviewer for his/her positive assessment of our article.